# Paper-Based Electrochemical Biosensors for Voltammetric Detection of miRNA Biomarkers Using Reduced Graphene Oxide or MoS_2_ Nanosheets Decorated with Gold Nanoparticle Electrodes

**DOI:** 10.3390/bios11070236

**Published:** 2021-07-13

**Authors:** Hilal Torul, Ece Yarali, Ece Eksin, Abhijit Ganguly, John Benson, Ugur Tamer, Pagona Papakonstantinou, Arzum Erdem

**Affiliations:** 1Department of Analytical Chemistry, Faculty of Pharmacy, Gazi University, Ankara 06330, Turkey; hilaltorul@gazi.edu.tr (H.T.); utamer@gazi.edu.tr (U.T.); 2Department of Analytical Chemistry, Faculty of Pharmacy, Ege University, Bornova 35100, Turkey; eceyarali@hotmail.com (E.Y.); eceksin@hotmail.com (E.E.); 3School of Engineering, Engineering Research Institute, Ulster University, Newtownabbey BT37 0QB, UK; a.ganguly@ulster.ac.uk; 42-DTech, Core Technology Facility, 46 Grafton Street, Manchester M13 9NT, UK; john.benson@2-dtech.com

**Keywords:** paper-based biosensor, reduced graphene oxide, molybdenum disulfide nanosheets, microRNA, gold nanohybrids, differential pulse voltammetry

## Abstract

Paper-based biosensors are considered simple and cost-efficient sensing platforms for analytical tests and diagnostics. Here, a paper-based electrochemical biosensor was developed for the rapid and sensitive detection of microRNAs (miRNA-155 and miRNA-21) related to early diagnosis of lung cancer. Hydrophobic barriers to creating electrode areas were manufactured by wax printing, whereas a three-electrode system was fabricated by a simple stencil approach. A carbon-based working electrode was modified using either reduced graphene oxide or molybdenum disulfide nanosheets modified with gold nanoparticle (AuNPs/RGO, AuNPs/MoS_2_) hybrid structures. The resulting paper-based biosensors offered sensitive detection of miRNA-155 and miRNA-21 by differential pulse voltammetry (DPV) in only 5.0 µL sample. The duration in our assay from the point of electrode modification to the final detection of miRNA was completed within only 35 min. The detection limits for miRNA-21 and miRNA-155 were found to be 12.0 and 25.7 nM for AuNPs/RGO and 51.6 and 59.6 nM for AuNPs/MoS_2_ sensors in the case of perfectly matched probe-target hybrids. These biosensors were found to be selective enough to distinguish the target miRNA in the presence of single-base mismatch miRNA or noncomplementary miRNA sequences.

## 1. Introduction

The use of paper in chemical analysis started as early as the 1930s [1,2], and the first paper-based glucose sensor was fabricated in the 1950s [3]. However, paper-based sensors were identified as a distinctive category by Whiteside et al. in 2007 [4,5]. In the past decade, paper-based sensors have received increased interest because they are easy to use and disposable with low-cost fabrication [6,7,8]. They also provide benefits, such as short analysis time and usage of a small volume of sample [9]. Therefore, they are promising alternatives to traditional point-of-care devices. A typical paper-based electrochemical sensor consists of a paper as a substrate material, an electrode area, and two or three electrodes. To fabricate the electrode area, hydrophobic barriers are prepared using several techniques, such as chemical vapor-phase deposition, soft lithography, wax patterning, and inkjet printing [5]. Two- or three-electrode systems can be fabricated using various techniques, such as photolithography [10], stencil printing [11], and inkjet printing [12]. Paper-based sensors find application in fields such as environmental analysis, biomedicine, food safety, chemical industry, and clinical analysis [13,14,15,16,17]. They can also be used as a type of point-of-care (POC) systems since they exhibit eco-friendly behavior [18,19,20,21]. 

Cancer is one of the most prevalent genetic diseases that leads to uncontrolled cell growth and deregulation of gene expression. miRNAs have been associated with cancer due to their wide impact on gene expression [22]. miRNAs have functions as oncogenes or tumor suppressors based on their inhibition of tumor-suppressive and oncogenic target mRNAs [23,24,25]. For example, miRNA-34, miRNA-126, miRNA-133, miRNA-143, and miRNA-145 are downregulated in many types of cancer. miRNA-15, miRNA-16, miRNA-21, miRNA-155, and miRNA-372 are found to be highly expressed in different types of tumors and promote oncogenesis [26]. Recent works have shown that miR-21 and miR-155 are the most significantly altered miRNAs in most cancer types [22,27,28]. Due to their critical roles in cancer and other diseases, miRNAs are considered crucial noninvasive biomarkers. Therefore, efforts have been made in recent years to develop innovative platforms for the efficient detection of microRNAs. Several methods, such as polymerase chain reaction (PCR)-based techniques, miRNA microarrays, sequencing, Northern blotting, mass spectrometry, optical and electrochemical methods have been examined for sensitive and selective miRNA detection [29,30,31,32,33].

Among them, electrochemical detection techniques have been considered as cost-effective and, at the same time, can provide high selectivity and sensitivity detection. Advances in electrochemical biomolecular detection are well discussed in a work reported by Sage et al. [34]. Paper-based sensors can be perfectly integrated with electrochemical techniques [35]. These combined systems enable the integration of different electrodes, such as ion-selective or microelectrodes [36,37]. They provide signal amplification while working in flow injection systems. Besides, paper-based analytical devices can also be coupled with several electrochemical methods, such as impedimetric detection, amperometry, cyclic voltammetry, coulometry, and potentiometric techniques [5].

In the present study, paper-based electrochemical biosensors based on gold nanoparticle hybrids were developed for selective and sensitive miRNA-155 and miRNA-21 detection. A carbon-based working electrode of a paper-based biosensor was modified using nanosheets of either reduced graphene oxide or molybdenum disulfide decorated with gold nanoparticles (denoted as AuNPs/RGO and AuNPs/MoS_2_, respectively). The sequence of a complementary miRNA target was detected by a thiol-linked synthetic DNA probe, immobilized onto the working electrode by differential pulse voltammetry (DPV) in a redox [Fe(CN)_6_]^3−/4−^ solution. According to the differentiation at the signal of redox probe measured in the absence/presence of miRNA hybridization on the paper-based biosensor, hybridization was detected.

## 2. Materials and Methods 

### 2.1. Apparatus

Electrochemical measurements were carried out with an Autolab-302 PGSTAT and GPES 4.9.007 software package (Eco Chemie, Utrecht, The Netherlands). A Faraday cage (Eco Chemie, Utrecht, The Netherlands) was used to reduce background signal. Raw data were treated with a Savitzky–Golay filter (level 2) and moving average baseline correction (peak width, 0.03).

### 2.2. Chemicals

Carbon paste was purchased from Daejoo Electronic Materials Co., LTD. (Siheung-si, Korea). Ag/AgCl (9:1) ink was purchased from Henkel (Dusseldorf, Germany). Reduced graphene oxide (RGO) was produced by 2-DTech Ltd. (Manchester, UK) using a proprietary approach. Nitrocellulose (NC) membrane (Hi-Flow Plus HFC07504) was provided by Merck (Darmstadt, Germany). *N*-hydroxysuccinimide (NHS), chloroauric acid (HAuCl_4_), and *N*-(3-dimethylaminopropyl)-*N*′-ethylcarbodiimide hydrochloride (EDC) were obtained from Sigma-Aldrich (St. Louis, MI, USA). The EDC/NHS solution was prepared at 10.0 mM concentration for each component in pH 7.4 phosphate buffer. miRNAs and the base sequences of all oligonucleotides are given in the Appendix A.

All other reagents were purchased from Sigma-Aldrich and Merck.

### 2.3. Generation and Modification of Paper Electrode

First, the paper electrode was developed as reported in our previous work [38]. It was constructed using a nitrocellulose membrane. After the construction of a pattern including a fluidic channel and electrode assembly area, a hydrophobic barrier was generated by utilizing a wax printer onto the NC membrane. Channels with a diameter of 2.0 mm and a length of 1.5 cm were constructed for capillary flow, and the length of the resulting channel was 0.6 cm. We placed three electrodes assembled in the working area designed with dimensions of ~20 mm^2^ and a 270 angle to obtain the maximum spread speed of the liquid. A pattern was designed onto a steel wafer of 0.1 mm thickness using a laser cutter. The resulting mask was placed on the NC membrane, and commercial carbon ink was used to create the working and counter electrodes. For the pseudo-reference electrode, an Ag/AgCl ink was used, and copper wires were used as conductive pads. The resulting electrode assembly was backed at 100 °C for 5 min. A schematic illustration of the electrode assembly is indicated in Scheme 1.

RGO powder was dispersed in ultrapure water at 1.0 mg/mL by a sonicator for 2 h. In order to obtain an RGO-modified paper electrode, 3.0 µL of 1.0 mg/mL RGO aqueous solution was applied on the surface of a working electrode three times. Between each drop of RGO aqueous solution, the electrode surface was dried under tungsten lamp for 5 min. 

Chemical activation was carried out with EDC/NHS as a cross-linking agent. Each RGO-modified paper electrode surface was covered with 5.0 μL of EDC/NHS solution and interacted for 20 min to activate carboxyl groups on the surface of the RGO-modified paper electrode.

### 2.4. Preparation of a Molybdenum Disulfide Nanosheet (MoS_2_)-Modified Paper Electrode 

In this study, few-layer MoS_2_ nanosheets were prepared by ionic-liquid-assisted grinding exfoliation, followed by sequential centrifugation steps, as described in our previous studies [39]. MoS_2_ powder was dispersed in ultrapure water at 2.0 mg/mL by a sonicator. The surface of the working electrode was covered by dropping 3.0 µL of MoS_2_ aqueous solution. Then, the electrode surface was dried under tungsten lamp for 5 min.

### 2.5. Electrodeposition of Gold Nanoparticles (AuNPs) on an RGO- or MoS_2_-Modified Paper Electrode 

After the preparation of an RGO- or MoS_2_-modified paper electrode, AuNPs were deposited onto the modified paper electrodes using the chronoamperometric technique in aqueous solution of a HAuCl_4_ gold precursor by applying −0.3 V for 10 min.

### 2.6. miRNA Detection with AuNP/RGO- or AuNP/MoS_2_-Modified Paper Electrodes

The surface of AuNP/RGO- or AuNP/MoS_2_-modified paper electrodes was covered with 5.0 μL of thiol-linked Probe-1 or Probe-2. The DNA probe was covalently immobilized onto AuNPs. Then, a washing step was applied using PBS (pH 7.4) to prevent nonspecific binding. The hybridization of the probe and target microRNAs was achieved by dropping 5.0 μL of miRNA-155, or miRNA-21, on the surface of the electrodes. After the hybridization step, the prepared electrodes were washed with PBS in order to eliminate nonspecific adsorption.

### 2.7. Voltammetric Measurement

The experiments were carried out in 20.0 μL of a 1.0 mM [Fe(CN)_6_]^3−/4−^ redox probe between −0.1 and +0.3 V at a scan rate of 50 mV/s and a pulse amplitude of 50 mV by DPV. 

CV measurements were performed by scanning between −1.0 and +1.0 V at a scan rate of 50 mV/s in a redox probe solution of 50.0 mM [Fe(CN)_6_]^3−/4−^ prepared in 0.1 M KCl. 

## 3. Results and Discussion

### 3.1. Characterization Studies of the Paper Electrode Modified with Gold Nanoparticles/Reduced Graphene Oxide (AuNPs/RGO)

The characterization of the modified paper electrodes was achieved by Raman spectroscopy. A Raman microscope (DeltaNu Inc., Laramie, WY, USA) with a charge-coupled device detector, a laser source at 785 nm, and a motorized XYZ microscope stage specimen holder was utilized to characterize the working electrode surface. The measurements were achieved by using a 10X objective with a laser spot size of 7.5 μm. Raman signals were obtained with a laser power of 140 mW for an acquisition time of 20 s.

Raman spectra of the RGO-modified paper electrodes are shown in Figure 1A. The Raman peak of RGO at 1308 cm^−1^ was attributed to the D band correlated with the structural defects or disorders in the lattice structure. The band at 1590 cm^−1^ was related to the G band associated with the first-order scattering of the E_2g_ vibrational mode [40,41]. Gold interfacing on RGO enhanced the intensity of the D and G bands by 79.2% and 78.7%, respectively. The enhancement of the signals can be via the excitation of localized surface plasmons or the formation of charge-transfer complexes between RGO and AuNPs [42].

The morphological characterization of RGO- and AuNP/RGO-modified paper electrodes was realized using a Quanta 200 3D scanning electron microscope (SEM). As shown in Figure 1B(b), the resulting AuNPs were homogeneously dispersed onto the RGO surface. The size of the gold nanoparticle was found to be 229 ± 53 nm and covered both RGO flakes and the working electrode area. These results demonstrate that AuNPs can be successfully electrodeposited onto agglomerates of RGO. A typical SEM image of RGO is shown in Figure 2, revealing a crumple-like morphology.

The electrochemical characterization of an unmodified paper electrode, RGO-modified paper electrode, and AuNP/RGO-modified paper electrode was performed by cyclic voltammetry (Appendix A). Identifying the anodic and cathodic current peaks occurring from the electrolysis of a redox-active solution, [Fe(CN)_6_]^3^^−/4^^−^, the anodic and cathodic current values (Ia and Ic) were estimated from the respective peak intensities, and the charges (Qa and Qc) were calculated from the area encapsulated under the respective peaks. The results are given in Appendix A for all types of electrodes.

The highest Ia and Ic were recorded by a AuNP/RGO-modified paper electrode (Appendix A). Substantial increase in both Ia and Ic, compared with an RGO-modified paper electrode, confirmed that the role of AuNPs is to enhance the electrode conductivity by facilitating the electron transfer [43,44,45].

The electroactive surface area (A) of each electrode—unmodified paper electrode, RGO-modified paper electrode, and AuNP-decorated RGO-modified paper electrode—was calculated by using the Randles–Sevcik equation [46] (Equation (1)), where Ip is the peak current (Ia or Ic) in A, n is the number of transferred electrons, A is the surface area in cm^2^, D is the diffusion coefficient in cm^2^/s, C is the concentration of electroactive species in mol/cm^3^, and v is the scan rate in V/s.
*Ip* = 2.687 × 10^5^ × n^3/2^ × A × D^1/2^ × C × v^1/2^(1)

The electroactive surface area of the paper electrodes was calculated based on Ia and found to be 0.020 cm^2^ for the unmodified paper electrode, 0.026 cm^2^ for the RGO-modified paper electrode, and 0.036 cm^2^ for the AuNP-decorated RGO-modified paper electrode (shown in Appendix A). An increase of about 80% was obtained at the electroactive surface area in the presence of a modification with AuNPs and RGO in comparison with the unmodified paper electrode due to the increase of the conductivity of the electrode based on the nature of the RGO nanomaterial and gold nanoparticles [47]. Furthermore, the AuNP/RGO-modified paper electrode exhibited about 39% increase in the electroactive surface area, confirming that the AuNP modification can enhance electroactivity, hence the sensitivity of the RGO-modified paper electrode.

### 3.2. Voltammetric Detection of miRNA-155 and miRNA-21 by a AuNP/RGO-Modified Paper Electrode

The detection of hybridization relies on the change of the oxidation signal of a redox [Fe(CN)_6_]^3−/4−^ probe. The immobilization of a thiol-linked DNA probe onto the electrode leads to a decrease in peak current. This result suggests that the hindrance is caused by the negatively charged DNA probe, while preventing the diffusion of the redox probe [Fe(CN)_6_]^3−/4−^ to the working electrode surface. The peak current was decreased after forming probe/miRNA target hybrids due to the presence of a more negatively charged DNA–miRNA hybrid at the electrode surface. The decrease at the peak current also indicates the forming perfect-match DNA probe/its complementary miRNA target hybrids [48].

All experiments for the detection of miRNA hybridization were efficiently carried out using a AuNP- and RGO-modified paper electrode to optimize the probe concentration, probe immobilization time, and hybridization time. The obtained results for the optimization studies are shown in Appendix A.

Hybridization efficiency (HE%) is calculated as evidence of the probe and miRNA hybridization efficacy in order to determine optimum conditions [49].

*HE%* = Δ*I × 100/I_probe_* represents the hybridization efficiency, where Δ*I = I_hybrid_ − I_probe_*. 

All experiments related to the detection of miRNA-155 and miRNA-21 were further explored under optimum conditions of this study.

After the optimization studies, the analytical performance of the electrodes was tested through the detection of a miRNA-155 target at different concentrations in the range of 0.25–2.0 µg/mL. Accordingly, the voltammograms regarding the oxidation signals are shown in Figure 3A,B. The highest HE% is calculated and found to be 37.1% in the case of Probe-1 and 1.0 µg/mL miRNA-155 target hybridization (see Appendix A).

The identical procedure was applied for voltammetric detection of miRNA-21, which is another biomarker of non-small-cell lung carcinoma (NSCLC). Similarly, the analytical performance of the electrodes was tested through the detection of a miRNA-21 target at different concentrations in the range of 0.25–2.0 µg/mL. Accordingly, the voltammograms are shown in Figure 3C,D. The highest HE% is calculated and found to be 43.2% in the case of Probe-2 with 1.0 µg/mL miRNA-21 target hybridization (see Appendix A).

The detection limit (LOD) [50] was calculated to be 0.19 µg/mL (25.71 nM, 128.0 fmol in 5.0 µL sample) for miRNA-155 via linear fitting of the calibration curve with the equation y = −10.63x + 27.05 and R^2^ = 0.98 (shown in Figure 4A). Similarly, the LOD of miRNA-21 was calculated to be 0.08 µg/mL (12.0 nM, 60.0 fmol in 5.0 µL sample) by fitting the calibration curve using the equation y = −12.64x + 30 and R^2^ = 0.99 (Figure 4B). Additionally, the sensor sensitivity was estimated from the slope of the calibration curve, divided by the surface area of the AuNP/RGO-paper electrode, for miRNA-155 and miRNA-21, and found to be 295.3 and 351.1 µA·mL/µg·cm^2^, respectively.

### 3.3. Selectivity of the Assay on the Voltammetric Detection of miRNA-155 by the AuNP/RGO-Modified Paper Electrode

The selectivity of the assay was then investigated against other miRNAs; a single-base mismatch (MM) or noncomplementary (NC) ones and the results are given in Appendix A. In the absence of the target sequence, the average oxidation signal of [Fe(CN)_6_]^3^^−/4^^−^ was measured to be 29.47 ± 0.44 µA. This signal decreased to 17.32 ± 3.22 µA (RSD%, 18.64%, *n* = 10) after occurring the perfect-match Probe-1 and its target miRNA-155 hybrids (Appendix A). On the other hand, the average signal was obtained as 20.05 ± 2.35 µA and 20.04 ± 2.71 µA in the case of hybridization between Probe-1 and NC or MM, respectively (Appendix A). However, the oxidation peak current of [Fe(CN)_6_]^3^^−/4^^−^ was measured to be 18.40 ± 1.62 µA and 18.15 ± 7.10 µA when hybridization was performed in the mixture samples consisting of target:NC (1:1) and target:MM (1:1), respectively (Appendix A). The highest decrease (i.e., 41.2%) at the oxidation signal of [Fe(CN)_6_]^3−/4−^ was obtained in the case of a full-match hybridization in contrast to the ones obtained by NC or MM sequences (Appendix A). Moreover, the standard deviations and RSD % values were high in the presence of NC or MM sequences due to the noneffective hybridization. Considering the number of bases that are similar to the target sequence (4 base pairing with NC, 22 base pairing with MM, see Appendix A), it is expected that the sensor developed exhibited a more selective behavior towards to the NC sequence than the MM sequence. In fact, the standard deviation and RSD% value obtained in the presence of NC were better than those obtained with MM. Hence, it can be concluded that the present assay offered a selective detection of miRNA even if the assay was examined in the mixture samples containing a miRNA target with other miRNA sequences, which differed one base from the target miRNA sequence or noncomplementary miRNA sequence (Appendix A).

### 3.4. Selectivity of the Assay on the Detection of miRNA-21 by Differential Pulse Voltammetry Using an AuNP/RGO-Modified Paper Electrode

The selectivity of the assay was then investigated against NC or MM (Appendix A). The average oxidation signal of [Fe(CN)_6_]^3−/4−^ was determined to be 17.00 ± 3.17 µA (RSD%, 18.65%, *n* = 2) after forming the perfect-match Probe-2 and miRNA-21 target hybrids (Appendix A), whereas the average signal was measured to be 20.64 ± 5.75 µA and 18.65 ± 4.12 µA after the hybridization of Probe-2 with NC and MM, respectively (Appendix A). Hence, it can be concluded that the present assay offered a selective behavior even if the assay was formed from the mixture of the miRNA target and the oligonucleotides, which differed one base from target miRNA sequence or noncomplementary miRNA sequence (Appendix A).

### 3.5. Characterization Studies of the Paper Electrode Modified with Gold Nanoparticle–Molybdenum Disulfide Nanosheets (AuNP/MoS_2_) 

The characterization of the AuNP- and MoS_2_-modified paper electrode was achieved by Raman spectroscopy under the conditions indicated previously. Raman signals were obtained for the characterization of modified paper electrodes (Figure 5A). Three main Raman peaks in the wave number range of 300–500 cm^−1^ correspond to MoS_2_ [51,52]. The peak at 381 cm^−1^ is attributed to the in-plane vibration of two S atoms and Mo (E^1^_2g_). The peak at 409 cm^−1^ is related to the out-plane vibration of S atoms (A^1^_g_). Another main MoS_2_ peak at 452 cm^−1^ is due to the 2 LA mode. The obtained Raman spectra proved the existence of MoS_2_ on the working electrode surface. As shown in Figure 5A, the SERS effect was observed after gold nanoparticle deposition on the modified surface. The signals of MoS_2_ molecules were increased by gold deposition. This result is also evidence of gold deposition onto the surface.

The MoS_2_-nanosheet- and AuNP/MoS_2_-modified paper electrodes were characterized by a Quanta 200 3D SEM. Figure 5B shows the deposition of bare MoS_2_ nanosheets on the carbon-ink-modified NC paper electrode. After electrodeposition, the gold nanoparticles can clearly be seen on the MoS_2_-modified paper electrode surface (Figure 5B). The diameter of AuNPs was measured to be 540 ± 140 nm. The SEM image in Figure 6 shows that the exfoliation process resulted in MoS_2_ nanosheets with lateral dimensions of ~1 µm and a wide range of smaller nanosheets stacked on the larger ones.

The electrochemical characterization of the unmodified paper electrode, MoS_2_-modified paper electrode, and AuNP deposition was performed by cyclic voltammetry (Appendix A). The charges (Qa and Qc) and currents (Ia (μA) and Ic (μA)) with the surface area of each electrode are shown in Appendix A.

The electroactive surface area (A) was calculated according to Ia and found to be 0.020 cm^2^ for the unmodified paper electrode, 0.021 cm^2^ for the MoS_2_-modified paper electrode, and 0.035 cm^2^ for the AuNP/MoS_2_-modified paper electrode (shown in Appendix A). After AuNP/MoS_2_ modification, the electroactive surface area of the AuNP/MoS_2_-modified paper electrode was increased by about 75% compared with the unmodified one by means of a layered structure of MoS_2_ nanosheets and the conductive nature of AuNPs.

### 3.6. Voltammetric Detection of miRNA-155 and miRNA-21 by the AuNP- and MoS_2_-Modified Paper Electrodes

All experiments were carried out by the AuNP- and MoS_2_-modified paper electrodes for the optimization of the developed method, such as probe immobilization time and hybridization time. The obtained results are shown in Appendix A. Further experiments on miRNA-21 and miRNA-155 detection were carried out under optimum conditions in the present study. 

The oxidation signals based on miRNA hybridization at different concentrations of miRNA-21 from 0.5 to 5.0 µg/mL were measured by the DPV technique. Figure 7A and Appendix A show the representative voltammograms with the resulting line graph. 

The LOD of miRNA-21 was calculated and found to be 0.36 µg/mL (51.68 nM, 258 fmol in 5.0 µL sample) using the equation y = −2.77x + 23.39 and R^2^ = 0.99 (shown in Figure 7B) by AuNP/MoS_2_-modified paper electrodes.

The calculated HE% values on hybridization with the miRNA-21 target are given in Appendix A.

Similarly, the oxidation signals of miRNA-155 hybridization were measured voltammetrically at different concentrations of miRNA-155 from 1.0 to 4.0 µg/mL. Appendix A shows the representative voltammograms with the line graph of the AuNP/MoS_2_-modified paper electrodes. The highest HE% was calculated and found to be 32% in the presence of hybridization with a 2.0 µg/mL miRNA-155 target (see Appendix A). The LOD [50] was also calculated and found to be 0.44 µg/mL (59.67 nM, 298 fmol in 5.0 µL sample) for miRNA-155 with the equation y = −4.71x + 28.15 and R^2^ = 0.97 (shown in Appendix A).

Additionally, the sensitivity of the AuNP/MoS_2_-modified paper electrode was estimated for miRNA-21 and miRNA-155 and found to be 79.1 and 134.6 µA·mL/µg·cm^2^, respectively.

### 3.7. Selectivity of the Assay on the Detection of miRNA-155 by Differential Pulse Voltammetry Using the AuNP- and MoS_2_-Modified Paper Electrodes

The selectivity of the assay was investigated against NC or MM (Appendix A). The average [Fe(CN)_6_]^3−/4−^ oxidation signal was determined to be 19.74 ± 1.75 µA (RSD%, 8.88%, *n* = 6) after forming the perfect-match Probe-1/miRNA-155 target hybrids (Appendix A), whereas the average signal was recorded to be 30.77 ± 8.37 µA and 22.52 ± 2.80 µA after the hybridization of Probe-1 with NC and MM, respectively (Appendix A). Moreover, the developed paper-electrode-based DNA probe could identify its complementary target miRNAs with high selectivity in the samples containing NC or MM by measuring nearly the same signal in contrast to the perfect-match hybridization signal (Appendix A). Hence, it can be concluded that the developed assay offered a selective behavior even if the assay was formed from the mixture of the miRNA target and the oligonucleotides, which differed one base from the target miRNA sequence or noncomplementary miRNA sequence.

### 3.8. Selectivity of the Assay on the Detection of miRNA-21 by Differential Pulse Voltammetry Using the AuNP- and MoS_2_-Modified Paper Electrodes

Similarly, the selectivity of the assay was investigated against NC or MM (Appendix A). The average [Fe(CN)_6_]^3−/4−^ oxidation signal was recorded to be 16.20 ± 3.12 µA (RSD%, 19.27%, *n* = 8) after occurring the perfect-match hybrid between Probe-2 and the miRNA-21 target (Appendix A), and there was a 29.73% decrease in comparison with the signal measured in the absence of the target. On the other hand, there were 17% and 6% increases and 6% and 5% decreases after the hybridization of Probe-2 with NC, MM, target:NC mixture, and target:MM mixture, respectively (Appendix A). Since, the highest decrease at the [Fe(CN)_6_]^3−/4−^ oxidation signal was obtained in the case of full-match hybridization in contrast to the ones obtained by NC or MM sequences, it can be concluded that the developed assay offered a selective behavior.

## 4. Conclusions

In this study, paper-based electrochemical biosensors were presented for sensitive detection of microRNA (i.e., miRNA-155 and miRNA-21) biomarkers related to early diagnosis of lung cancer for the first time. Hydrophobic barriers to creating electrode areas were constructed by wax printing, whereas the three-electrode system was fabricated by simple mask printing. The surface of the working electrode was modified using either gold-nanoparticle-reduced graphene oxide or gold-nanoparticle–molybdenum disulfide nanosheets. The electroactive surface areas of AuNP/RGO and AuNP/MoS_2_-modified paper electrodes (about 80% and 75%, respectively) were increased with respect to unmodified ones. The resulting paper-based biosensors exhibited good reproducibility by the incorporation of unique properties of RGO and MoS_2_ nanosheets. Additionally, AuNPs played an excellent role in the signal amplification.

Here, the voltammetric analysis of miRNA-155 and miRNA-21 resulted in a relatively shorter detection time in comparison with earlier studies related to biosensors (Table 1). The entire assay performed at room temperature, including electrode modification and miRNA detection, was completed in 35 min. A single droplet (5.0 µL) of a sample was enough to cover the entire working electrode area, which enabled analysis in low sample volumes. Barring a few exceptions, the sample volumes used in previous works are in the range of 5–100 µL. Therefore, the sample volume of our assay is one of the lowest volumes among the studies summarized in Table 1. The LODs of miRNA-21 were calculated to be 12.00 and 51.68 nM using a AuNP/RGO-modified paper electrode and a AuNP/MoS_2_-modified paper electrode, respectively. On the other hand, the LODs of miRNA-155 were found to be 25.71 and 59.67 nM using a AuNP/RGO-modified paper electrode and a AuNP/MoS_2_-modified paper electrode, respectively. In contrast to the results obtained by the AuNP/MoS_2_-modified paper electrode, the AuNP/RGO-modified paper electrode performed miRNA detection with more sensitive results. Overall, the studies indicate that our proposed assay with nanosheet-modified paper electrodes detected miRNA hybridization accurately in contrast to one-base mismatch miRNA or noncomplementary miRNA. The proposed assay offers some advantages over earlier reports on miRNA detection (summarized in Table 1) in terms of ease of use, short assay time (35 min), and low cost per analysis. Additionally, it is important to note that our method simplifies the miRNA detection assay by avoiding the complex chemistries (i.e., cleaning of the electrode surface, formation of a self-assembled monolayer, usage of a nanoparticle-attached DNA probe) in sensor fabrication steps in comparison to earlier reports [53,54].

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
