# Peer review of "Paper-Based Electrochemical Biosensors for Voltammetric Detection of miRNA Biomarkers Using Reduced Graphene Oxide or MoS2 Nanosheets Decorated with Gold Nanoparticle Electrodes"

_biosensors, 2021, doi:10.3390/bios11070236_

Round 1

Reviewer 1 Report

Dear Editor,

The manuscript entitled “Paper-based electrochemical biosensors for voltammetric detection of miRNA biomarkers using reduced graphene oxide or MoS2 nanosheets decorated with gold nanoparticle electrodes” by Torul, Yarali, Ece Eksin et al. presents the development of paper-based electrochemical biosensors for detection of miRNA-155 and miRNA-21, based on reduced graphene oxide or molybdenum disulfide nanosheets hybrid structures modified with gold nanoparticles, as carbon-based working electrodes. The authors utilize differential pulse voltammetry (DPV) for detection of miRNA-155 and miRNA-21 with 5 µL of sample, in short time (35 min). The detection limits for miRNA-21 and miRNA-155 was found to be 12.0 nM and 25.7 nM for AuNPs/RGO, and 51.6 nM and 59.6 nM for AuNPs/MoS2 sensors, respectively. The biosensors were tested for selectivity against single base mismatch miRNA or non-complementary miRNA sequences.

In my opinion, the manuscripts’ objective and perspective are very interesting, the manuscript is well-written and should be accepted for publication after minor revisions. My detailed comments for the authors to consider are provided below:

  1. A graphic describing the assay steps and the detection principle would be useful for the reader.
  2. Figures 1 and 2 should be reversed in order. I understand why the authors preferred to present the RGO SEM first (figure 1), and the paper-based tests as figure 2, however figure 2 is discussed before figure 1 and this was a bit confusing. The same applies to figures 5 and 6.
  3. The letters in equation 1 should be explained. Also, Ia in paragraph 1, page 6 is the Ip referred to equation 1?
  4. Throughout the text minor spelling mistakes and words duplication must be corrected.

Overall, it is a very well written and informative manuscript, in my opinion.

Author Response

RESPONSE TO REVIEWERS

Thanks for your comments regarding our manuscript (ID: biosensors-1260161, “Paper-based electrochemical biosensors for voltammetric detection of miRNA biomarkers using reduced graphene oxide or MoS2 nanosheets decorated with gold nanoparticle electrodes”). Enclosed you will find the revised manuscript and our detailed answers to the comments made by the three referees.

We hope our answers and revisions will now satisfy the referees and you will reconsider our manuscript for publication in Biosensors.

Thank you in advance.

Sincerely

# REVIEWER 1

The manuscript entitled “Paper-based electrochemical biosensors for voltammetric detection of miRNA biomarkers using reduced graphene oxide or MoS2 nanosheets decorated with gold nanoparticle electrodes” by Torul, Yarali, Ece Eksin et al. presents the development of paper-based electrochemical biosensors for detection of miRNA-155 and miRNA-21, based on reduced graphene oxide or molybdenum disulfide nanosheets hybrid structures modified with gold nanoparticles, as carbon-based working electrodes. The authors utilize differential pulse voltammetry (DPV) for detection of miRNA-155 and miRNA-21 with 5 µL of sample, in short time (35 min). The detection limits for miRNA-21 and miRNA-155 was found to be 12.0 nM and 25.7 nM for AuNPs/RGO, and 51.6 nM and 59.6 nM for AuNPs/MoS2 sensors, respectively. The biosensors were tested for selectivity against single base mismatch miRNA or non-complementary miRNA sequences.

In my opinion, the manuscripts’ objective and perspective are very interesting, the manuscript is well-written and should be accepted for publication after minor revisions. My detailed comments for the authors to consider are provided below:

Referee’s comment: A graphic describing the assay steps and the detection principle would be useful for the reader.

Response: A schematic illustration of the Probe/miRNA assembling steps has been prepared and provided in the supplementary material at page 3.

Referee’s comment: Figures 1 and 2 should be reversed in order. I understand why the authors preferred to present the RGO SEM first (figure 1), and the paper-based tests as figure 2, however figure 2 is discussed before figure 1 and this was a bit confusing. The same applies to figures 5 and 6.

Response: Figures 1 and 2 have been reversed in order as the referee has requested and we have also applied the same change for figures 5 and 6.

Referee’s comment: The letters in equation 1 should be explained. Also, Ia in paragraph 1, page 6 is the Ip referred to equation 1?

Response: The letters in Randles-Sevcik equation have been explained at page 6, as the referee has requested. The referee is right that Ia has been used as Ip in the equation 1.

Referee’s comment: Throughout the text minor spelling mistakes and words duplication must be corrected.

Response: The manuscript is thoroughly checked, and all the necessary corrections are done in the entire text.

Overall, it is a very well written and informative manuscript, in my opinion.

Reviewer 2 Report

Torul et al report on a paper-based electrochemical biosensor for detection of micro RNA biomarkers related to early diagnosis of lung cancer. The working electrode was modified with AuNPs/RGO and AuNPs/MoS2 structures. Morphological, chemical and electrochemical characterization of the device surface and the biomarkers immobilization are presented. The authors demonstrate an improvement of the biosensor response after following a strategy based on Au NPs on RGO and MoS2 nanosheets. The authors also performed a selectivity study using single-base mismatch and non-complementary miRNA. Overall, this article may be accepted after minor revision. Please, consider the following comments below:

Please, check the manuscript entirely for typos, misplaced comma and mistaken punctuation

1) I suggest some improvements on Scheme 1. The authors can reorganize the elements for better arrangement of the items displayed. Additionally, a schematic illustration of the Probe/micro RNA assembling could be incorporated to help readers to understand the device functionalization steps.

2) The authors have compared the performance of their biosensors with other similar ones. I suggest including the concentration range of biosensors on Table 1 for a complete comparison. Additionally, the authors say they have developed a sensitive biosensor. For this affirmation, the authors should compare their sensitivity value with other works considering the concentrations of microRNA found in real samples.

3) For a good correlation on a calibration curve, please report the detection of 4 or 5 different analyte concentrations at least. (for example, see Figure 8A where only 3 different concentration have been measured).

4) In both reported biosensors, the same DNA Probe was used to detect each type of micro RNA. Can authors explain the difference of performance between AuNPs/RGO and AuNPs/MoS2-modified paper electrodes?

5) I am not completely convinced on the selectivity of the biossensor AuNP-RGO shown on Figure S7, S8 and S12. The figure caption is also a bit confuse. Can authors explain better the data?

Author Response

RESPONSE TO REVIEWERS

Thanks for your comments regarding our manuscript (ID: biosensors-1260161, “Paper-based electrochemical biosensors for voltammetric detection of miRNA biomarkers using reduced graphene oxide or MoS2 nanosheets decorated with gold nanoparticle electrodes”). Enclosed you will find the revised manuscript and our detailed answers to the comments made by the three referees.

We hope our answers and revisions will now satisfy the referees and you will reconsider our manuscript for publication in Biosensors.

Thank you in advance.

Sincerely

# REVIEWER 2

Torul et al report on a paper-based electrochemical biosensor for detection of micro RNA biomarkers related to early diagnosis of lung cancer. The working electrode was modified with AuNPs/RGO and AuNPs/MoS2 structures. Morphological, chemical and electrochemical characterization of the device surface and the biomarkers immobilization are presented. The authors demonstrate an improvement of the biosensor response after following a strategy based on Au NPs on RGO and MoS2 nanosheets. The authors also performed a selectivity study using single-base mismatch and non-complementary miRNA. Overall, this article may be accepted after minor revision. Please, consider the following comments below:

Referee’s comment: Please, check the manuscript entirely for typos, misplaced comma and mistaken punctuation.

Response: The manuscript is thoroughly checked, and all the necessary corrections are done in the entire text.

Referee’s comment: I suggest some improvements on Scheme 1. The authors can reorganize the elements for better arrangement of the items displayed. Additionally, a schematic illustration of the Probe/micro RNA assembling could be incorporated to help readers to understand the device functionalization steps.

Response: “Scheme 1” has been reorganized as the referee has requested and provided in the revised manuscript at page 4. We have also prepared a schematic illustration of the Probe/miRNA assembling and provided in the supplementary material at page 3.

Referee’s comment: The authors have compared the performance of their biosensors with other similar ones. I suggest including the concentration range of biosensors on Table 1 for a complete comparison. Additionally, the authors say they have developed a sensitive biosensor. For this affirmation, the authors should compare their sensitivity value with other works considering the concentrations of microRNA found in real samples.

Response: The concentration range and the sample volumes of current miRNA biosensors are included on Table 1. The revised version of Table 1 is given below. As can be seen from Table 1, our assay offers very short assay time (35 min) in comparison to earlier reports on miRNA detection. A single droplet (5 µL) of sample was enough for analysis, which means miRNA analysis can be performed in very low sample volumes. Additionally, our method simplifies the miRNA detection assay by avoiding the complex chemistries (i.e. cleaning of the electrode surface, formation of self-assembled monolayer, usage of nanoparticle attached DNA probe etc.) in sensor fabrication steps compared to earlier reports. This benefit makes our assay cost-effective as well.

The benchmark in terms of features mentioned above is included in the conclusion section.

Additionally, the sensor sensitivity values were estimated from the slope of the calibration curve divided by the surface area of paper electrodes as follows,

“The sensor sensitivity was estimated from the slope of the calibration curve, divided by the surface area of AuNPs/RGO-paper electrode, for both miRNA-155 and miRNA-21 and found to be 295.3 µA.mL/µg.cm2 and 351.1 µA.mL/µg.cm2, respectively.”

“Additionally, the sensitivity of AuNPs/MoS2- paper electrode was estimated for miRNA-21 and miRNA-155, and found to be 79.1 µA.mL/µg.cm2 and 134.6 µA.mL/µg.cm2, respectively.”

Table 1. Comparison of different electrochemical biosensors for the detection of miRNA-155 and miRNA-21.

Abbreviations: MWCNT: multiwalled carbon nanotube, GCE: glassy carbon electrode, AuNRs: Gold nanorods, AuE: gold electrode, ITO: indium tin oxide, GONRs: graphene oxide nanoribbons, SPGE: screen printed gold electrode, DPV: differential pulse voltammetry, CC: chronocoulometry, SWV: square wave voltammetry, ASV: striping voltammetry, EIS: Electrochemical impedance spectroscopy.

miRNA

Electrode

Method

Analysis Time

Sample volume

Concentration range

DL

Reference

miRNA-107

SPGE

DPV

75 min

30 µL

5 fM−5 pM

10 fM

[52]

Au-NPFe2O3NC/SPCE

CC

45 min

-

100 aM-1 nM

100 aM

[53]

miRNA-21

MoS2/Thi /AuNPs

nanocomposite/GCE

SWV

18 h

5 µL

1 pM-10 nM

0.26 pM

[54]

Au@NPFe2O3NC/GCE

CC

-

-

100 fM–1 µM

100 fM

[51]

AuNPs/ITO

ASV

17 h

100 µL

2.5 fM-25 nM

0.12 fM

[55]

AuNPs/GCE

DPV

3.5 h

40 µL

100 aM-1 nM

78.0 aM

[56]

AuNPs@MoS2/GCE

DPV

EIS

17 h

20 µL

10 fM-1 nM

0.78 fM

0.45 fM

[57]

MWCNTs@GONRs/AuNPs/GCE

DPV

14 h

6 µL

0.1 fM-0.1 nM

0.034 fM

[58]

AuE

DPV

14 h

2 µL

0.1 fM-1 nM

0.04 fM

[59]

AuNPs@MoS2/SPGE

AuNPs@/SPGE

CC

22 h

10 µL

100 aM-1 pM

10 fM-10 pM

100.0 aM

10.0 fM

[46]

AuNPs/RGO/PE

DPV

35 min

5 µL

37.5 nM-150 nM

12 nM

This study

AuNPs/MoS2/PE

143.5 nM-287.1 nM

51.7 nM

miRNA-155

AuE

SWV

8 h

-

0.5 pM-0.1 µM

0.13 fM

[60]

GO/Au/GCE

CV, DPV

21 h

10 µL

0.8 fM-1 nM

0.37 fM

[61]

nano-Pd/Thi/GCE

CV

17 h

20 µL

5.6 pM-5.6 µM

1.87 pM

[62]

AuNRs/GO/GCE

DPV

4 h

5 µL

2 fM-8pM

0.6 fM

[63]

AuNPs/RGO/PE

DPV

35 min

5 µL

33.8 nM-135.3 nM

25.7 nM

This study

AuNPs/MoS2/PE

33.9 nM-406.8 nM

59.7 nM

Referee’s comment: For a good correlation on a calibration curve, please report the detection of 4 or 5 different analyte concentrations at least. (for example, see Figure 8A where only 3 different concentration have been measured).

Response: Referee is right that the calibration curve should be created by using 4 or 5 different analyte concentrations at least. By presenting our apologies, we would like to indicate that we cannot find any chance to provide the detection of 4 or 5 different analyte concentrations because we carried out all experiments before a certain time. For this reason, we moved the Figure 8A and related figures regarding  Probe 1 and miRNA-155 to the supplementary material.

Referee’s comment: In both reported biosensors, the same DNA Probe was used to detect each type of microRNA. Can authors explain the difference of performance between AuNPs/RGO and AuNPs/MoS2-modified paper electrodes?

Response: For each of microRNA detection, different DNA probe was used. Thiol link miRNA-155 specific DNA probe, which was nominated as Probe-1, was used for the detection of miRNA-155. However, thiol link miRNA-21 specific DNA probe, which was nominated as Probe-2, was used for the detection of miRNA-21. The base sequences of Probe-1 and Probe-2 is given below. miRNAs and the base sequences of all oligonucleotides were listed in supporting information as well.

thiol link miRNA-155 specific DNA probe (Probe-1)

5’- SH-ACC CCT ATC ACG ATT AGC ATT AA-3’

thiol link miRNA-21 specific DNA probe (Probe-2)

5’- SH-TCA ACA TCA GTC TGA TAA GCT A-3’

The performance of both the electrodes, AuNPs/RGO and AuNPs/MoS2-modified paper electrodes, was given in the revised conclusion section of our manuscript as follows:

“In this study, paper-based electrochemical biosensors were presented for sensitive detection of microRNA (i.e. miRNA-155 and miRNA-21) biomarkers related to early diagnosis of lung cancer, for the first time. The hydrophobic barriers to create electrode areas were constructed by wax printing, whereas the three-electrode system was fabricated by simple mask printing. The surface of working electrode was modified using either gold nanoparticle-reduced graphene oxide, or gold nanoparticle-molybdenum disulfide nanosheets. The electroactive surface areas of AuNPs/RGO and AuNPs/MoS2-modified paper electrodes (about 80% and 75%, respectively) were increased with respect to unmodified ones. The resulting paper-based biosensors exhibited good reproducibility by incorporation of unique properties of RGO and MoS2 nanosheets. Additionally, AuNPs played an excellent role at the signal amplification.

The voltammetric analysis of miRNA-155 and miRNA-21 resulted herein in a relatively shorter detection time in comparison to earlier studies related to biosensors (Table 1). The entire assay performed at room temperature including electrode modification and miRNA detection was completed in 35 minutes. A single droplet (5.0 µL) of sample was enough to cover entire the working electrode area which enabled analysis in low sample volumes. Barring a few exceptions, sample volumes used in previous works are in the range of 5-100 µL. Therefore, the lowest sample volume is used in our study of all (Table 1). The LODs of miRNA-21 were calculated as 12.00 nM and 51.68 nM using AuNPs/RGO-modified paper electrode and AuNPs/MoS2-modified paper electrode, respectively. On the other hand, LODs of miRNA-155 were found as 25.71 nM and 59.67 nM using AuNPs/RGO-modified paper electrode and AuNPs/MoS2-modified paper electrode, respectively. In contrast to the results obtained by AuNPs/MoS2-modified paper electrode, the AuNPs/RGO-modified paper electrode performed miRNA detection with more sensitive results. Overall, the studies indicated that our proposed assay with nanosheets modified paper electrodes detected miRNA hybridization accurately in contrast to one base mismatch miRNA or non-complementary miRNA. The proposed assay offers some advantages over to earlier reports on miRNA detection (summarized in Table 1) in terms of ease to use, short assay time (35 min), low cost per analysis. Additionally, it is important to note that our method simplifies the miRNA detection assay by avoiding the complex chemistries (i.e. cleaning of the electrode surface, formation of self-assembled monolayer, usage of nanoparticle attached DNA probe etc.) in sensor fabrication steps compared to earlier reports [50,51].”

Referee’s comment: I am not completely convinced on the selectivity of the biosensor AuNP-RGO shown on Figure S7, S8 and S12. The figure caption is also a bit confuse. Can authors explain better the data?

Response: In Figure S7, voltammograms representing the [Fe(CN)6]3-/4- oxidation signal obtained by Probe-1 immobilized AuNPs/RGO-paper electrode, hybridization between probe and miRNA-155, non-complementary (NC), single-base mismatched strand (MM), and mixture samples with miRNA-155 and NC or MM. The average [Fe(CN)6]3-/4- oxidation signals measured before and after hybridization of      Probe-1 with miRNA-155 target, NC, MM, the mixture sample containing target:NC (1:1) or target:MM (1:1) were listed in Table S5.

The same study performed in order to test the selectivity of the assay to miRNA-21 and the results were given in Figure S8. In this case, Probe-2 immobilized onto the surface of AuNPs/RGO-paper electrode, and hybridization was performed with target, NC, MM, target:NC mixture, target:MM mixture sample. In Figure S8, voltammograms representing the [Fe(CN)6]3-/4- oxidation signal obtained by Probe-1 immobilized AuNPs/RGO-paper electrode, hybridization between probe and miRNA-21, non-complementary (NC), single-base mismatched strand (MM), and mixture samples with miRNA-21 and NC or MM. The average [Fe(CN)6]3-/4- oxidation signals measured before and after hybridization of Probe-2 with miRNA-21 target, NC, MM, the mixture sample containing target:NC (1:1) or target:MM (1:1) were listed in Table S6.

The same selectivity test was performed with AuNPs/MoS2-modified paper electrode and the results were given as Figure S15 and Figure S16. The average [Fe(CN)6]3-/4- oxidation signals and HE% calculated according to the oxidation signals obtained after hybridization were listed in table S10 and Table S11, as well.

In order to avoid confusion, figure and table captions are revised. The revised form of each Figure and Table are given below.

Figure S7. (A) Voltammograms representing the [Fe(CN)6]3-/4- oxidation signal obtained by (a) Probe-1 immobilized AuNPs/RGO-modified paper electrode in the absence of miRNA-155 target, after hybridization of Probe-1 with (b) miRNA-155 target, (c) NC, and (d) MM, individually.  (B) Voltammograms representing the [Fe(CN)6]3-/4- oxidation signal obtained by (a) Probe-1 immobilized AuNPs/RGO-modified paper electrode in the absence of miRNA-155 target, after hybridization of Probe-1 (b) with only miRNA-155 target, (c) in target:NC (1:1) mixture, and (d) in target:MM (1:1) mixture.

Table S5. The average [Fe(CN)6]3-/4-  oxidation signals measured before and after hybridization of Probe-1 with miRNA-155 target, NC, MM, the mixture sample containing target:NC (1:1) or the mixture sample containing target:MM (1:1). HE% calculated according to the oxidation signals obtained after hybridization.  

Figure S8. (A) Voltammograms representing the [Fe(CN)6]3-/4- oxidation signal obtained by (a) Probe-2 immobilized AuNPs/RGO-modified paper electrode in the absence of miRNA-21 target, after hybridization of Probe-2 with (b) miRNA-21 target, (c) NC, and (d) MM, individually.  (B) Voltammograms representing the [Fe(CN)6]3-/4- oxidation signal obtained by (a’) Probe-2 immobilized AuNPs/RGO-modified paper electrode in the absence of miRNA-21 target, after hybridization of Probe-2 (b’) with only miRNA-21 target, (c’) in target:NC (1:1) mixture, and (d’) in target:MM (1:1) mixture.

Table S6. The average [Fe(CN)6]3-/4-  oxidation signals measured before and after hybridization of Probe-2 with miRNA-21 target, NC, MM, the mixture sample containing target:NC (1:1) or the mixture sample containing target:MM (1:1). HE% calculated according to the oxidation signals obtained after hybridization. 

Figure S15. (A) Voltammograms representing the [Fe(CN)6]3-/4- oxidation signal obtained by (a) Probe-1 immobilized AuNPs/MoS2-modified paper electrode in the absence of miRNA-155 target, after hybridization of Probe-1 with (b) miRNA-155 target, (c) NC, and (d) MM, individually. (B) Voltammograms representing the [Fe(CN)6]3-/4- oxidation signal obtained by (a) Probe-1 immobilized AuNPs/MoS2-modified paper electrode in the absence of miRNA-155 target, after hybridization of Probe-1 (b) with only miRNA-155 target, (c) in target:NC (1:1) mixture, and (d) in target:MM (1:1) mixture.

Table S10. The average [Fe(CN)6]3-/4- oxidation signals (n=2) measured before and after hybridization of Probe-1 with miRNA-155 target, NC, MM, the mixture sample containing target:NC (1:1) or the mixture sample containing target:MM (1:1). HE% calculated according to the oxidation signals obtained after hybridization. 

Figure S16. (A) Voltammograms representing the [Fe(CN)6]3-/4- oxidation signal obtained by (a) Probe-2 immobilized AuNPs/MoS2-modified paper electrode in the absence of miRNA-21 target, after hybridization of Probe-2 with, (b) miRNA-21 target, (c) NC, and (d) MM, individually.  (B) Voltammograms representing the [Fe(CN)6]3-/4- oxidation signal obtained by (a) Probe-2 immobilized AuNPs/MoS2-modified paper electrode in the absence of miRNA-21 target, after hybridization of Probe-2 (b) with only miRNA-21 target, (c) in target:NC (1:1) mixture, and (d) in target:MM (1:1) mixture.

Table S11. The average [Fe(CN)6]3-/4- oxidation signals (n=2) measured before and after hybridization of Probe-2 with miRNA-21 target, NC, MM, the mixture sample containing target:NC (1:1) or the mixture sample containing target:MM (1:1). HE% calculated according to the oxidation signals obtained after hybridization.

Reviewer 3 Report

In this manuscript the authors are reporting the fabrication of RGO/AuNPs and MoS2/AuNPs paper-based electrodes that are further used as electrochemical biosensors for miRNA biomarkers detection.

The English has to be thoroughly improved so that a proper revision of the manuscript to be conducted.

Moreover, I suggest the authors a thoroughly revision of the text to be made by a scientist with experience in publishing in electrochemistry field before submitting it for publication.

I am presenting here only few examples of unproperly use of specific terms:

Lines 143-144: „The hybridization of paper-based probe and biomarkers was achieved by dropping 5.0 μL of miRNA-155, or miRNA-21 on the surface of probe immobilized electrodes.“

Line 198: „An increase about 80% was obtained at the electroactive surface area in the presence of modification with AuNPs and RGO in comparison to the unmodified paper electrode due to the increase at the conductivity of the electrode based on the nature of the RGO nanomaterial and gold nanoparticles“

Line 217:“ The hybridization efficiency (HE%) is calculated as the signature of the probe and miRNA hybridization effectiveness“

Line 224: „„the voltammograms regarding the oxidation signals were shown in ...“

Line 228: „the oxidation signals of DNA:miRNA hybridization occured in six different concentrations of 0.25-2.0 μg/mL of miRNA-21 target were followed by DPV“;

Line 250: „The detection limit (LOD) [47] was reported as 0.19 μg/... in a concentration range of 0-1.0 μg/mL“

The caption of figures are also unproperly described/formulated: „Figure 4. (A) The calibration curve regarding the average [Fe(CN)6]3-/4- oxidation signal (n=3) created after nucleic acid hybridization“

As well, some explanation or presented data are not clear enough: e.g. Line 184: what are the authors meaning by „relative charges (Qa and Qc)“; Line 189: „the decreased slope after gold deposition is indicative of enhanced electron transfer ability“

The calculation of the calibration curves and of the related data is not clear. The sensor performance seems quite low.

The paper can’t be published in this form as it lacks in too many aspects.

Author Response

RESPONSE TO REVIEWERS

Thanks for your comments regarding our manuscript (ID: biosensors-1260161, “Paper-based electrochemical biosensors for voltammetric detection of miRNA biomarkers using reduced graphene oxide or MoS2 nanosheets decorated with gold nanoparticle electrodes”). Enclosed you will find the revised manuscript and our detailed answers to the comments made by the three referees.

We hope our answers and revisions will now satisfy the referees and you will reconsider our manuscript for publication in Biosensors.

Thank you in advance.

Sincerely

# REVIEWER 3

In this manuscript the authors are reporting the fabrication of RGO/AuNPs and MoS2/AuNPs paper-based electrodes that are further used as electrochemical biosensors for miRNA biomarkers detection.

The English has to be thoroughly improved so that a proper revision of the manuscript to be conducted. Moreover, I suggest the authors a thoroughly revision of the text to be made by a scientist with experience in publishing in electrochemistry field before submitting it for publication. I am presenting here only few examples of unproperly use of specific terms:

Lines 143-144: “The hybridization of paper-based probe and biomarkers was achieved by dropping 5.0 μL of miRNA-155, or miRNA-21 on the surface of probe immobilized electrodes.”

Line 198: “An increase about 80% was obtained at the electroactive surface area in the presence of modification with AuNPs and RGO in comparison to the unmodified paper electrode due to the increase at the conductivity of the electrode based on the nature of the RGO nanomaterial and gold nanoparticles”

Line 217: “The hybridization efficiency (HE%) is calculated as the signature of the probe and miRNA hybridization effectiveness”

Line 224: “the voltammograms regarding the oxidation signals were shown in ...”

Line 228: “the oxidation signals of DNA:miRNA hybridization occured in six different concentrations of 0.25-2.0 μg/mL of miRNA-21 target were followed by DPV”

Line 250: “The detection limit (LOD) [47] was reported as 0.19 μg/... in a concentration range of 0-1.0 μg/mL”

Response: The manuscript is thoroughly checked, and all the necessary corrections are done in the entire text.

Referee’s comment: The caption of figures are also unproperly described/formulated: „Figure 4. (A) The calibration curve regarding the average [Fe(CN)6]3-/4- oxidation signal (n=3) created after nucleic acid hybridization“

Response: All the captions are revised to avoid confusion. The revised versions of Figure captions are given below.

Figure 1. (A) Raman spectra of (a) nitrocellulose membrane (black), (b) carbon paste (red), (c) RGO-modified paper electrode (blue), (d) RGO-modified paper electrode after AuNPs deposition (pink), (B) SEM images of (a) RGO, (b) AuNPs/RGO-modified paper electrodes (Scale: 5 µm). Gold deposition was performed with chronoamperometry technique in aqueous solution of HAuCl4 gold precursor, by applying -0.3 V, during 10 min.

Figure 2. SEM image of RGO (Scale: 5 µm)

Figure 3. (A) Voltammograms representing the [Fe(CN)6]3-/4- oxidation signals obtained by (a) Probe-1 immobilized AuNPs/RGO-paper electrode, after hybridization of Probe-1 with miRNA-155 target at the concentrations of (b) 0.25 µg/mL, (c) 0.5 µg/mL, (d) 0.75 µg/mL, (e) 1.0 µg/mL, (f) 1.5 µg/mL, (g) 2.0 µg/mL. (B) The line graph based on the average [Fe(CN)6]3-/4- oxidation signal after hybridization between Probe-1 and miRNA-155 target with its various concentrations from 0 to 2.0 µg/mL (n=3). (C) Voltammograms representing the [Fe(CN)6]3-/4- oxidation signals obtained by (a) Probe-2 immobilized AuNPs/RGO-paper electrode after hybridization of Probe-2 with miRNA-21 target at  the concentrations of (b) 0.25 µg/mL, (c) 0.5 µg/mL, (d) 0.75 µg/mL, (e) 1.0 µg/mL, (f) 1.5 µg/mL, (g) 2.0 µg/mL. (D) The line graph based on the average [Fe(CN)6]3-/4- oxidation signal measured after hybridization between Probe-2 and miRNA-21 target with its various concentrations from 0 to 2.0 µg/mL (n=3).

Figure 4. (A) The calibration plot, for AuNPs/RGO-paper electrode, obtained after hybridization between Probe-1 and miRNA-155 target with its various concentrations from 0 to 1.0 µg/mL (n=3). (B) The calibration plot for the same electrode obtained after hybridization between Probe-2 and miRNA-21 target with its various concentrations from 0 to 1.0 µg/mL (n=3).

Figure 5. (A) Raman spectra of paper electrode. (a) nitrocellulose membrane (black), (b) carbon paste (red), (c) MoS2-modified paper electrode (blue), (d) MoS2-modified paper electrode after AuNPs deposition (red). (B) SEM images of (a) MoS2 nanosheets-modified paper electrode, (b) MoS2-modified paper electrode after gold electrodeposition (Scale: 2 µm).

Figure 6. SEM image of MoS2 nanosheets (Scale: 2 µm).

Figure 7. (A) Voltammograms representing the [Fe(CN)6]3-/4- oxidation signals obtained by (a) Probe-2 immobilized AuNPs/MoS2- paper electrode after hybridization of Probe-2 with miRNA-21 target at  the concentrations of (b) 0.5 µg/mL, (c) 1.0 µg/mL,  (d) 1.5 µg/mL, (e) 2.0 µg/mL, (f) 2.5 µg/mL, (g) 3.0 µg/mL, (h) 5.0 µg/mL. (B) The calibration plot, for AuNPs/MoS2-paper electrode, obtained after hybridization between Probe-2 and miRNA-21 target with its various concentrations from 0 to 3.0 µg/mL (n=3).

Referee’s comment: As well, some explanation or presented data are not clear enough: e.g. Line 184: what are the authors meaning by “relative charges (Qa and Qc)”; Line 189: “the decreased slope after gold deposition is indicative of enhanced electron transfer ability”

Response:  According to your suggestion, explanation for CV data is revised as follows,

“Identifying the anodic and cathodic current peaks, occurring from the electrolysis of redox active solution, [Fe(CN)6]3-/4-, the anodic and cathodic current values (Ia and Ic) were estimated from the respective peak intensities, as well as the charges (Qa and Qc) were calculated from the area encapsulated under the respective peaks. The results are given in Table S1 for all type of electrodes.”

Line 189 is deleted.

Referee’s comment: The calculation of the calibration curves and of the related data is not clear. The sensor performance seems quite low.

Response: The statistical method described by Miller and Miller which is based on the S/N ratio is used to evaluate limit of detection. More information about calculation method can be found in Statistics and Chemometrics for Analytical Chemistry by Miller & Miller (Miller, J.N.; Miller, J.C. Statistics and Chemometrics for Analytical Chemistry; 6th ed.; Pearson: Harlow, UK, 2010; ISBN 0273730428, 9780273730422.)

Round 2

Reviewer 3 Report

The manuscript has been reviewed and much improved. However there are still a few aspects that should be clarified before publication.

  1. I am still recommending the authors to reanalyse the Introduction section. For example: line 64: „The advances in electrochemical biomolecular detection are well discussed in the work reported by Sage et al. [31]. They are also known as simple portable instruments.“ – the second sentence it is not suited in the context.
  2. Page 9, Section 3.3 and Page 10 Section 3.4 – the current oxidation values for [Fe(CN)6]3-/4- should be presented in relation with the value in absence of the target. However the data suggest that the assay is not very selective, especially in regard with the single-base mismatch. The RSD are quite high. The authors should mention this aspect.
  3. Conclusions: „Therefore, the lowest sample volume is used in our study of all“ – the sentence is not clear
  4. Supplementary file, page 5 - „The hybridization of 0.5 µg/mL Probe-1 was immobilized onto the surface of electrode“ –the sentence is not correct

Supplementary file, Page 5 - „In the absence of Probe-1 the oxidation signal was measured as 32.96 ± 5.12 µA using AuNPs/RGO-modified paper electrode. There was 1.6% and 22 % decrease at the oxidation signal after“ – page 5. Here it is not clear what the authors are presenting. Is the signal measured after hybridization with the target, or directky after the probe immobilization? Please mention that. What is the reason they have chosen these times 10 and 30 min and didn’t try higher adsorption times? The authors should introduce a short mention about these choices

Supplementary file, Page 6 – „The highest decrease at the oxidation signal was obtained as 28.9% in case of 5 min hybridization time.“ Same question: did the authors tried lower or higher hybridization time? Another suggestion is to present next to these percent values the refence, i.e. 28.9% from the current registered in the target absence.

Author Response

RESPONSE TO REVIEWER

We hope the last revision will now satisfy the referee and you will reconsider our manuscript for publication in Biosensors.

Thank you in advance.

Sincerely

# REVIEWER 3

The manuscript has been reviewed and much improved. However, there are still a few aspects that should be clarified before publication.

Referee’s comment: I am still recommending the authors to reanalyse the Introduction section. For example: line 64: “The advances in electrochemical biomolecular detection are well discussed in the work reported by Sage et al. [31]. They are also known as simple portable instruments.” – the second sentence it is not suited in the context.

Response: The introduction section was checked thoroughly, and significant changes were made.

Referee’s comment: Page 9, Section 3.3 and Page 10 Section 3.4 – the current oxidation values for [Fe(CN)6]3-/4- should be presented in relation with the value in absence of the target. However, the data suggest that the assay is not very selective, especially in regard with the single-base mismatch. The RSD are quite high. The authors should mention this aspect.

Response: According to Reviewer’s suggestion, oxidation signal of [Fe(CN)6]3-/4- presented in relation with the value in absence of the target. All the signal values with standard deviations were given in Table S5 as below. Besides, we apologize since we previously made an unacceptable mistake by using high RDS values. We now checked all data and recalculated the currents with lower RSD values. In addition, the  discussion on details were included too.

Table S5. The average [Fe(CN)6]3-/4- oxidation signals measured before and after hybridization of Probe-1 with miRNA-155 target, NC, MM, the mixture sample containing target:NC (1:1) or the mixture sample containing target:MM (1:1). HE% calculated according to the oxidation signals obtained after hybridization. 

I (µA)

Probe-1 immobilized

AuNPs/RGO-modified paper electrode

29.47 ± 0.44 µA

HE%

miRNA-155 target

17.32 ± 3.22 µA

41.2%

NC

20.05 ± 2.35 µA 

31.9%

MM

20.04 ± 2.71 µA

31.9%

miRNA-155 target:NC (1:1) mixture

18.40 ± 1.62 µA 

37.5%

miRNA-155 target:MM (1:1) mixture

21.67 ± 5.12 µA

26.4%

Additionally, in order to show hybridization effectiveness, HE % values were calculated and given in Table S5. The hybridization efficiency (HE%) is calculated as an evidence of the probe and miRNA hybridization efficacy in order to determine optimum conditions [49].

HE% = D I x 100 / I probe represents the hybridization efficiency, where DI = Ihybrid – I probe

The highest HE % (i.e. 41.2%) was obtained in case of full match hybridization in contrast to the ones obtained by NC or MM sequences (Table S5).

The standard deviations and RSD % values were high in the presence of NC or MM sequences due to the non-effective hybridization. Considering the number of bases that are similar to the target sequence (4 base pairing with NC, 22 base pairing with MM, see supporting information), it is expected that the sensor developed exhibited more selective behavior towards the NC sequence than the MM sequence. In fact, the standard deviation and RSD % value obtained in the presence of NC are better than those obtained with MM.

More discussion is added related to the selectivity as follows;

“The selectivity of the assay was then investigated against to other miRNAs; single-base mismatch (MM) or non-complementary (NC) ones and the results were given in Figure S7. In the absence of the target sequence the average oxidation signal of [Fe(CN)6]3-/4- was measured as 29.47 ± 0.44 µA. This signal decreased to 17.32 ± 3.22 µA (RSD%, 18.64%, n=10) after occurring the perfect match Probe-1 and its target miRNA-155 hybrids (Figure S7A-b). On the other hand, the average signal was obtained as 20.05 ± 2.35 µA and 20.04 ± 2.71 µA in the case of hybridization between Probe-1 and NC or MM, respectively (Figure S7A-c, Figure S7A-d). However, the oxidation peak current of [Fe(CN)6]3-/4- was measured as 18.40 ± 1.62 µA  and  18.15 ± 7.10 µA when hybridization performed in the mixture samples consisted of target:NC (1:1), or target:MM (1:1), respectively (Figure S7B-c, Figure S7B-d). The highest decrease (i.e. 41.2 %) at oxidation signal of [Fe(CN)6]3-/4- was obtained in case of full match hybridization in contrast to the ones obtained by NC or MM sequences (Table S5). Moreover, the standard deviations and RSD % values were high in the presence of NC or MM sequences due to the non-effective hybridization. Considering the number of bases that are similar to the target sequence (4 base pairing with NC, 22 base pairing with MM, see supporting information), it is expected that the sensor developed exhibited more selective behavior towards the NC sequence than the MM sequence. In fact, the standard deviation and RSD % value obtained in the presence of NC are better than those obtained with MM. Hence, it could be concluded that the present assay offered a selective detection of miRNA even if the assay was examined in the mixture samples containing miRNA target with other miRNA sequences which differ one-base from target miRNA sequence, or non-complementary miRNA sequence (Table S5).”

Referee’s comment: Conclusions: „Therefore, the lowest sample volume is used in our study of all“ – the sentence is not clear

Response: The required correction is done. Revised version of the sentence is as follows: “Therefore, the sample volume of our assay is one of the lowest volumes among the studies summarized in Table 1”

Referee’s comment: Supplementary file, page 5 - „The hybridization of 0.5 µg/mL Probe-1 was immobilized onto the surface of electrode“ –the sentence is not correct

Response: The required correction is done as follows:

“In the absence of Probe-1, the oxidation signal of [Fe(CN)6]3-/4- was measured as 32.96 ± 5.12 µA by AuNPs/RGO-modified paper electrode. The average oxidation signal of [Fe(CN)6]3-/4- was measured as 32.43 ± 1.12 µA (RSD%,  3.47%, n=3) and 25.60 ± 0.14 µA (RSD%,  0.55%, n=3) after immobilization of 0.5 µg/mL Probe-1 during 10 and 30 min, respectively (Figure S5). According to the signal measured in the absence of Probe-1, the highest decrease at the oxidation signal was obtained in the case of 30 min Probe-1 immobilization as 22 % (Table S2). Thus, 30 min was chosen as optimum probe immobilization time for our further studies.”

Referee’s comment: Supplementary file, Page 5 - „In the absence of Probe-1 the oxidation signal was measured as 32.96 ± 5.12 µA using AuNPs/RGO-modified paper electrode. There was 1.6% and 22 % decrease at the oxidation signal after“ – page 5. Here it is not clear what the authors are presenting. Is the signal measured after hybridization with the target, or directky after the probe immobilization? Please mention that. What is the reason they have chosen these times 10 and 30 min and didn’t try higher adsorption times? The authors should introduce a short mention about these choices

Response: The required corrections are done in the whole passage and the revised version is given below;

“The effect of probe immobilization time onto the electrode surface upon the hybridization process

For the optimization of probe immobilization time, a short period (i.e. 10 min) which was used in our previous work [35] in contrast to a longer period (i.e. 30 min) were tested. Since the electrode surface dried over 30 min, there is no need to examine a much longer immobilization time in our study.

“In the absence of Probe-1, the oxidation signal of [Fe(CN)6]3-/4- was measured as 32.96 ± 5.12 µA by AuNPs/RGO-modified paper electrode. The average oxidation signal of [Fe(CN)6]3-/4- was measured as 32.43 ± 1.12 µA (RSD%,  3.47%, n=3) and 25.60 ± 0.14 µA (RSD%,  0.55%, n=3) after immobilization of 0.5 µg/mL Probe-1 during 10 and 30 min, respectively (Figure S5). According to the signal measured in the absence of Probe-1, the highest decrease at the oxidation signal was obtained in the case of 30 min Probe-1 immobilization as 22 % (Table S2). Thus, 30 min was chosen as optimum probe immobilization time for our further studies.”

Referee’s comment: Supplementary file, Page 6 – „The highest decrease at the oxidation signal was obtained as 28.9% in case of 5 min hybridization time“. Same question: did the authors tried lower or higher hybridization time? Another suggestion is to present next to these percent values the refence, i.e. 28.9% from the current registered in the target absence.

Response: The required corrections are done in the whole passage and the revised version is given below;

“The effect of hybridization time upon the hybridization process

For the optimization of hybridization time, a shorter hybridization time (i.e. 5 min) which was used in our previous work [35], in contrast to a longer period  (i.e. 15 min) were tested.

The hybridization of 0.5 µg/mL Probe-1 and 1.0 µg/mL miRNA-155 target was done during 5 and 15 min (Figure S6). The average oxidation signal of [Fe(CN)6]3-/4- was measured as 16.70 ± 4.17 µA (RSD%, 24.98%, n=4)  and 20.36 ± 10.68 µA (RSD%, 52.46%, n=4) after hybridization of Probe-1 with miRNA-155 target during 5 min and 15 min, respectively. According to the signal measured in the absence of target, the highest decrease (28.9% ) was recorded in the case of 5 min hybridization time. Thus, it was chosen as optimum hybridization time for further studies.”

Additionally, the reference signal is added next to the percentage value throughout the text.
